# Comparison of Different Machine Learning Algorithms for the Prediction of the Wheat Grain Filling Stage Using RGB Images

**DOI:** 10.3390/plants12234043

**Published:** 2023-11-30

**Authors:** Yunlin Song, Zhuangzhuang Sun, Ruinan Zhang, Haijiang Min, Qing Li, Jian Cai, Xiao Wang, Qin Zhou, Dong Jiang

**Affiliations:** National Technique Innovation Center for Regional Wheat Production, Key Laboratory of Crop Ecophysiology, Ministry of Agriculture, Nanjing Agricultural University, Nanjing 210095, China; 2018201011@njau.edu.cn (Y.S.); 2020201018@stu.njau.edu.cn (Z.S.); 2020801253@stu.njau.edu.cn (R.Z.); 2023101025@stu.njau.edu.cn (H.M.); caijian@njau.edu.cn (J.C.); xiaowang@njau.edu.cn (X.W.); qinzhou@njau.edu.cn (Q.Z.)

**Keywords:** wheat, grain filling, RGB image, machine learning, deep learning, few-shot learning

## Abstract

Grain filling is essential for wheat yield formation, but is very susceptible to environmental stresses, such as high temperatures, especially in the context of global climate change. Grain RGB images include rich color, shape, and texture information, which can explicitly reveal the dynamics of grain filling. However, it is still challenging to further quantitatively predict the days after anthesis (DAA) from grain RGB images to monitor grain development. Results: The WheatGrain dataset revealed dynamic changes in color, shape, and texture traits during grain development. To predict the DAA from RGB images of wheat grains, we tested the performance of traditional machine learning, deep learning, and few-shot learning on this dataset. The results showed that Random Forest (RF) had the best accuracy of the traditional machine learning algorithms, but it was far less accurate than all deep learning algorithms. The precision and recall of the deep learning classification model using Vision Transformer (ViT) were the highest, 99.03% and 99.00%, respectively. In addition, few-shot learning could realize fine-grained image recognition for wheat grains, and it had a higher accuracy and recall rate in the case of 5-shot, which were 96.86% and 96.67%, respectively. Materials and Methods: In this work, we proposed a complete wheat grain dataset, WheatGrain, which covers thousands of wheat grain images from 6 DAA to 39 DAA, which can characterize the complete dynamics of grain development. At the same time, we built different algorithms to predict the DAA, including traditional machine learning, deep learning, and few-shot learning, in this dataset, and evaluated the performance of all models. Conclusions: To obtain wheat grain filling dynamics promptly, this study proposed an RGB dataset for the whole growth period of grain development. In addition, detailed comparisons were conducted between traditional machine learning, deep learning, and few-shot learning, which provided the possibility of recognizing the DAA of the grain timely. These results revealed that the ViT could improve the performance of deep learning in predicting the DAA, while few-shot learning could reduce the need for a number of datasets. This work provides a new approach to monitoring wheat grain filling dynamics, and it is beneficial for disaster prevention and improvement of wheat production.

## 1. Introduction

Wheat is the principal supply of energy for mankind, which highlights the fact that wheat yield plays a crucial role in world food security [1]. The grain filling stage is a crucial stage for wheat grain formation and maturation after anthesis. Therefore, grain filling lays the foundation for wheat yield formation, and the characteristics of grain filling determine the final grain weight, a major yield component of wheat [2].

However, grain filling is highly susceptible to adverse factors such as high temperatures, and these are more severe in the context of global climate change. As a result, for every 1 °C increase in the maximum temperature during the wheat growing season, global wheat production decreases by 5.6% [3]. Some studies have also shown that grain filling characteristics include two important factors: the grain filling rate and grain filling duration [4], and their interaction determines the final grain weight of wheat [5]. In addition, other studies have shown that post-anthesis high-temperature stress could accelerate the grain filling rate and decrease the grain filling duration. Still, the increase in the grain filling rate is not enough to compensate for the loss caused by the shortening of the grain filling duration [6]. Therefore, the timely improvement of the grain filling process has guiding significance for the safety of wheat production in terms of disaster prevention and reduction.

The development of the grain shape has an important influence on the wheat grain filling process. The grain length, grain width, and grain thickness are crucial factors in determining the grain shape of wheat. Traditionally, these parameters were measured manually using vernier calipers [7]. However, this method was time-consuming, labor-intensive, prone to human error, and provided results that may not be representative due to the small sample size. RGB images of the grain can provide sufficient color, shape, and texture information [8], which has built a bridge between different inspections such as seed quality and purity. ImageJ 1.3.7 software can be used to recognize RGB images and extract more comprehensive grain morphology parameters. However, the grains should be arranged neatly for accurate identification, and the number of samples that can be processed simultaneously may be limited [9]. Some studies used computer vision technology to extract the texture and color traits of wheat, barley, oats, and rye, which classified each crop with a classification accuracy of 100% [10]. Similarly, four deep convolutional neural network models were proposed for the classification of lightly sprouted wheat kernels and sound wheat kernels, with over 95% accuracy on test sets [11]. By extracting 55 traits of wheat grains, Neethirajan et al. identified healthy and damaged wheat grains with an accuracy of 95% [12]. Although these studies have achieved progress in purity detection, viability detection, and the pest and disease detection of mature seeds, there is still a challenge in predicting the days after anthesis (DAA) of wheat grain due to the lack of an RGB image dataset for the complete grain filling stage.

A machine learning algorithm is an effective way to predict the DAA. Traditional machine learning algorithms, such as Decision Trees (DTs), Support Vector Machines (SVMs), and Random Forest (RF), have achieved satisfactory accuracy for wheat yield prediction [13], leaf area classification [14], and variety classification [15], although the accuracy of traditional machine learning algorithms depends on effective feature selection. However, manual feature selection is complicated, requiring a lot of optimization time and labor. In recent years, deep learning has become the mainstream in the field of computer vision, which can avoid feature engineering based on prior knowledge and achieve end-to-end classification or regression tasks. For example, a two-to-two deep learning model was designed to predict both wheat yield and quality from time-series data without any manual feature selection [16]. Furthermore, a CNN-LSTM framework has also provided evidence of the benefits of deep learning classification over traditional image analysis features for the classification of various plant genotypes [17]. Many studies have proved that deep learning algorithms can achieve satisfactory accuracy based on a large number of training samples, but some agriculture images are very time-consuming and laborious to obtain. To solve the data dependency problem in deep learning, few-shot learning was proposed, which has the ability to learn from a small number of training samples [18]. For example, Liang used few-shot learning to achieve the classification of cotton leaf spots, with the DenseNet algorithm having the best classification accuracy [19]. Similarly, a semi-supervised few-shot learning approach was proposed to recognize plant leaf diseases [20]. Although few-shot learning has many applications in plant disease classification, it is still unclear whether it can predict the wheat grain DAA or not, which requires fine-grained image recognition.

Wheat is one of the three major crops in the world [21], and the accurate prediction of the DAA from grain RGB images is of great significance for wheat development. Therefore, the highlights of this paper are: (1) A novel dataset that was established, with thousands of RGB images, which cover the period from 6 days to 39 days after anthesis; (2) we deciphered the dynamic changes in grain color, shape, and texture, during the grain filling stage, which revealed that RGB images can explicitly show the the dynamics of grain filling; and (3) we analyzed the performance of traditional machine learning, deep learning, and few-shot learning in predicting the DAA.

## 2. Results

### 2.1. Grain Filling Dynamics of Color, Shape, and Texture Traits

The color, shape, and texture traits changed during the grain filling process. For color traits, the values of R, G, and B were 243–252, 243–252, and 232–250, respectively, while the values of H, S, and V were 1–4, 2–18, and 243–253, respectively (Figure 1). In addition, R, G, B, and V showed a trend of first decreasing and then increasing with the DAA. At the same time, H and S showed a trend of first increasing and then decreasing.

For shape traits, the area, perimeter, radius, and equivalent diameter ranged from 9–32 mm^2^, 13–30 mm, 4–10 mm, and 3–6 mm, and the eccentric, compact, rectangle degree, and roundness ranged from 0.54–0.94, 16–40, 0.54–0.81, and 0.31–0.75, respectively (Figure 2). Furthermore, most of these traits showed a trend of first increasing and then decreasing, such as the area, perimeter, radius, and equivalent diameter. For example, the grain area increased sharply over days 6–24, slowly increased or was unchanged over days 24–33, and then declined after 33 days.

For texture traits, the homogeneity, dissimilarity, and correlation of the grain ranged from 0.25–0.73, 1–4, and 0.972–0.997, while the entropy, ASM, and energy ranged from 3.3–6.4, 0.001–0.062, and 0.03–0.025, respectively (Figure 3). Moreover, the homogeneity, correlation, ASM, and energy tended to decline with the increase in the DAA, while dissimilarity and entropy tended to rise with the increase in the DAA.

### 2.2. Machine Learning Performance for Predicting DAA

To verify whether machine learning can predict the DAA, three machine learning algorithms, including Decision Trees (DTs), Support Vector Machines (SVMs), and Random Forest (RF), were applied to predict the DAA. The results showed that the performance of the three machine learning algorithms was ranked as RF > SVM > DT (Figure 4). The average precision and recall of the DT, SVM, and RF were 76.11% and 74.83%, 80.98% and 80.78%, and 88.71% and 87.93%, respectively. In addition, the precision and recall of the three algorithms were higher at 6–15 days and 39 days, while the precision and recall were lower at 21–33 days.

### 2.3. Deep Learning Performance for Predicting DAA

To verify whether deep learning can predict DAA, an image classification model was constructed with different backbone networks, including HRnet18, Densenet121, Inceptionv3, VGG16, Resnet18, Resnet50, Resnet101, and Vision Transformer (ViT) (Figure 5 and Figure 6). The results showed that the image classification model with a ViT backbone network had the best performance, and its precision and recall were 99.03% and 99.00%, respectively (Table 1). On the other hand, the image classification model with a VGG16 backbone network had the worst performance, and its prediction precision and recall were 94.24% and 94.00%, respectively. However, the precision and recall of VGG16 also far outperformed that of the optimal machine learning algorithm, which had a precision and recall of 83.94% and 83.88%. In addition, all deep learning models also demonstrated a good prediction performance during the middle of the grain filling stage (21–33 days), compared with the machine learning performances. These results illustrated the advantages of deep learning algorithms not only in end-to-end image classification but also in providing higher accuracy.

### 2.4. Few-Shot Learning Performance for Predicting DAA

In order to reduce our dependence on a number of datasets, we further explored the performance of few-shot learning at predicting DAA. The results showed that the mean precision and mean recall of 1-shot were 92.23% and 91.67%, respectively. In addition, the precision and recall increased as the number of query sets increased. After 5-shot, the gain in the model’s performance started to become minuscule. In the end, the precision and recall of 5-shot were 96.86% and 96.67%, which was the optimal shot number (Figure 7).

## 3. Materials and Methods

### 3.1. Plant Material and Growth Conditions

The widely cultivated wheat varieties (*Triticum aestivum* L.) Xumai 32, Yangmai 15, and Yangmai 16 were planted at the Yanhu Farm in Tongshan District, Xuzhou City, Jiangsu Province (117°07′ E, 34°28′ N). The plot size was 3.5 m × 5 m, and the seedling density was 2.4 × 10^6^ ha^−1^ with a row space of 0.25 m. Two nitrogen (N) rates (225 and 270 kg N ha^−1^) were tested; the ratio of basal fertilizer (applied at the seedling stage) to topdressing fertilizer (applied at the jointing stage) was 5:5, respectively. All treatments were given 90 kg P_2_O_5_ ha^−1^ and 150 kg k_2_O ha^−1^ in the form of superphosphate and potassium chloride fertilizers before sowing. Other field management was conducted following local practices.

### 3.2. RGB Image Acquisition and Processing

In order to obtain RGB images of the 6–39 days of the grain filling period, wheat spike samples were taken from the field every three days, which began at six days after anthesis. After each sampling, 20 wheat grains were manually stripped from the middle of a spike. Twenty grains were neatly placed on a square black background plate for the photographing of each with a mobile phone (iPhone SE, Apple, Cupertino, CA, USA). A fixed device was also used to ensure that each shot was of the same focal length and lighting condition. The resolution of the obtained image was 3024 × 4032, and the image format was JPG. Then, a crop tool was used to crop to the image to the square black background plate with a 2000 × 2000 resolution. Finally, an object detection algorithm was used to detect the location of each grain with a 250 × 250 bounding box (Figure 8).

### 3.3. Dataset Construction

In order to compare the performance of different algorithms, the datasets need to be built separately, as required due to their characteristics. Deep learning and few-shot learning belong to an end-to-end model that can directly input raw RGB images, and convolutional neural networks (CNN) are used to automatically extract features; for traditional machine learning, it is necessary to manually carry out feature engineering, which determines the upper limit of the model.

#### 3.3.1. Dataset for Traditional Machine Learning

Machine learning algorithms require manual feature engineering. Therefore, the OTSU method [22] was used for each grain image to remove the black background and obtain a mask of the region of interest. A total of 20 color, shape, and texture traits were extracted for each grain. Specifically, six color traits are the means of R, G, B, H, S, and V; the eight shape traits include the area, perimeter, radius, equivalent diameter, eccentric, compact, rectangle degree, and roundness; the six texture traits include dissimilarity, homogeneity, energy, correlation, ASM, and entropy. These traits were extracted via Python and OpenCV (a Python library), and the codes are available online at https://github.com/shem123456/wheat-grain-traits (accessed on 21 September 2023).

#### 3.3.2. Dataset for Deep Learning

Deep learning is an end-to-end model structure, which can directly use raw RGB images as its input. The coordinate information of each grain is obtained based on the bounding boxes. A crop tool was used to crop the original images and save each grain as a 250 × 250 resolution image. These images were categorized and grouped into different folders according to their DAA (Appendix A). Finally, these images were randomly split into the training dataset, validation dataset, and testing dataset at proportions of 70%, 10%, and 20%, respectively.

#### 3.3.3. Dataset for Few-Shot Learning

Few-shot learning can learn the image mappings and distance representations from the source domain, so it can classify new categories with few images. Images from 80% of the deep learning dataset were used to fine-tune the pre-training weight of the Omniglot dataset [23]. The remaining 20% of the images were further divided into a query dataset (20%) and a testing dataset (80%). In addition, the typical structure of few-shot learning (N-way and K-shot) means that there are N classes with K samples per class in the target domain. There are 12 classes in this paper, which represent the 6–39 days of the grain filling period. The K is set to 1, 2, 5, and 10, which means that 1, 2, 5, and 10 images are randomly selected from the query dataset.

### 3.4. Model Construction

For traditional machine learning, DT, SVM, and RF were selected to predict the DAA. A decision tree is a tree structure (it can be a binary tree or a non-binary tree). Each non-leaf node represents a test for a feature attribute; each branch represents the output of this feature attribute over a range; and each leaf node stores a category. Based on the structural risk minimization criterion, SVM constructs the optimal classification hyperplane to maximize the classification interval to improve the generalization ability of the learning machine. The RF is an integrated Bagging-based learning method that can handle classification and regression [24]. All machine learning algorithms were built using Python and Scikit-Learn (a Python library).

For deep learning, convolutional neural networks are used to map one input to another output, which can automatically learn mapping relationships between massive volumes of data without manual feature engineering. The performance of the deep learning depends on the structure of the convolutional neural network, except for the Transformer model, which uses a multi-head attention mechanism. Therefore, HRnet18, Densenet121, Inceptionv3, VGG16, Resnet18, Resnet50, Resnet101, and ViT, were selected in this study, and these models were then evaluated to select the optimal network structure. All models are built using PyTorch (a deep learning framework) and timm (PyTorch Image Models: a collection of image models, layers, utilities, optimizers, schedulers, and data loaders), and trained on a high-performance computer with an Intel Xeon(R) Platinum 8350C CPU, 32 GB memory, and an NVIDIA 3090 GPU. The batch size was 32. The Adam method with a learning rate of 0.001 was set to optimize the model parameters.

For few-shot learning, a Siamese network with contrastive loss was used. There are two subnets in the Siamese network, and each subnet uses VGG16 as its backbone architecture. Meanwhile, both the subnets share the same model weight. When training, a pair of images is fed into the Siamese network, and two images will be mapped to feature vectors with the same dimensions. Then, the contrast loss is applied to calculate the similarity of the two feature vectors. If they belong to the same class, the network will minimize the loss function to shorten the distance between similar classes; otherwise, the network maximizes the loss function to distinguish between the different classes. The details of the Siamese architecture can be found in [25]. The definition of the contrastive loss function is shown in Formula (1). Finally, the Siamese network with contrastive loss was built using PyTorch, and the configuration of its training was consistent with that of the deep learning model described above. The codes are available online at https://github.com/shem123456/grain-filling-classification (accessed on 21 September 2023).
(1)$$Loss=yd^{2}+\left(1-y\right)\cdot max{\left(0,\mathrm{margin-}d\right)}^{2}$$
where *d* represents the Euclidean distance between the two samples and *y* is the label of whether the two samples match; *y* = 1 means that the two samples are the same class, and *y* = 0 means that the two samples are of different classes. The margin is the threshold.

### 3.5. Model Evaluation

We simplified the DAA prediction as a multiclass classification problem, so precision and recall, as common metrics for image classification, were applied to evaluate the models’ performance. Precision is the proportion of correct positive predictions to all positive predictions, while recall is the proportion of correct positive predictions to all positive examples. The relevant calculations are provided as follows:(2)$$precision=\frac{TP}{TP+FP}$$
(3)$$recall=\frac{TP}{TP+FN}$$
where *TP* represents true positives; *FP* represents false positives; and *FN* represents false negatives.

## 4. Discussion

The traits from RGB images can reflect grain filling dynamics. Previous studies have shown that wheat grain filling follows a slow–fast–slow pattern due to the reactions for protein and starch synthesis [2,26]. Specifically, the development of the wheat grain can be divided into three stages: 0-15 days, 15–27 days, and 27 days to maturation. During the first stage, the grain fresh weight continuously increases because of its division and expansion. Then, the grain dry weight will increase further because of starch and protein accumulation at 15–27 days. Finally, the grain filling rate begins to decline after 27 days, and the grain fresh weight drops rapidly after 33 days due to desiccation [27]. As shown in Figure 2, the grain filling pattern is similar to the trend in shape traits, including the area, perimeter, and equivalent diameter of the grain, suggesting that the shape traits from RGB images can characterize the dynamics of grain filling. Meanwhile, the color and texture traits did not show a similar trend, but these traits improved information abundance for predicting the DAA based on the performance results of traditional machine learning.

The prediction performance of RF is optimal in terms of machine learning, as seen in Figure 4, but the accuracy of traditional machine learning is still limited by feature engineering. Compared with other machine learning methods, RF is an ensemble learning method and is robust to noise and outliers [28]. However, the prediction performance of deep learning is much greater than that of traditional machine learning. The main source of this difference is that the precision and recall of machine learning algorithms were lower at days 21–33 (Figure 4). This might be because the wheat grains at 21–33 days are too similar for the machine learning algorithms to classify. On the one hand, the upper limit of the model for machine learning depends on feature engineering, and although a total of 20 color, shape, and texture traits were extracted in this study, these manually selected traits still have limitations in representing seed properties. On the other hand, the input for deep learning is the complete RGB image, as it is an end-to-end image classification model, and the network structure of the CNN involved can select the optimal feature based on the gradient [29]. Notably, ViT achieved optimal prediction accuracy (Figure 5 and Figure 6), although it uses a network structure that is different from a CNN. ViT is the network construction of an attention mechanism [30], which can be a model that focuses on more useful information while suppressing redundancy, and this structure and its variants have generated state-of-the-art results in many visual tasks [31].

Few-shot learning can also achieve a better performance than traditional machine learning in the prediction of DAA. To explore whether few-shot learning can achieve fine-grained recognition for wheat grains, we evaluated the performance of a Siamese network with contrastive loss in predicting the DAA. As shown in Figure 7, the results showed that the few-shot learning under 1-shot could approximate the performance of a standard deep learning algorithm. Meanwhile, its mean precision and recall increased with the number of shots, which were consistent with other results for disease recognition [18]. These results demonstrate that few-shot learning is an effective strategy for reducing our dependence on the dataset scale, which benefits from metric-based learning to compare and analyze unseen samples [32].

Although a complete wheat grain filling dataset was presented, with a comparison of different algorithms, in this study, there are still some perspectives that need to be considered in the future: (1) Although the deep learning algorithm has great advantages in predictive performance, it is far from practical application in the field. In our next study, we will try some lightweight improvements to make it more suitable for mobile deployment. (2) In this study, only RGB images were used, while multispectral and hyperspectral imaging are also worth exploring to further obtain physiological traits.

## 5. Conclusions

Predicting the wheat grain DAA provides timely information on wheat grain filling dynamics, which is essential for ensuring the security of the wheat yield. In this work, we presented a complete wheat grain dataset that characterized the dynamic development of wheat grains. This dataset revealed the dynamic changes in traits such as color, shape, and texture during wheat grain development. To predict the DAA from RGB images of wheat grains, we tested the performance of traditional machine learning, deep learning, and few-shot learning on this dataset. The results showed that the deep learning classification model with ViT as the backbone neural network had the highest accuracy and recall, 99.03% and 99.00%, respectively, while few-shot learning had a reduced dependence on the dataset’s scale, and had a higher precision and recall in the case of 5-shot learning, 95.08% and 94.83%, respectively. In future work, we expect to propose a grain filling monitoring model compatible with few samples and which has high accuracy in order to achieve its production purpose: disaster prevention and mitigation.

## Figures and Tables

**Figure 1 plants-12-04043-f001:**
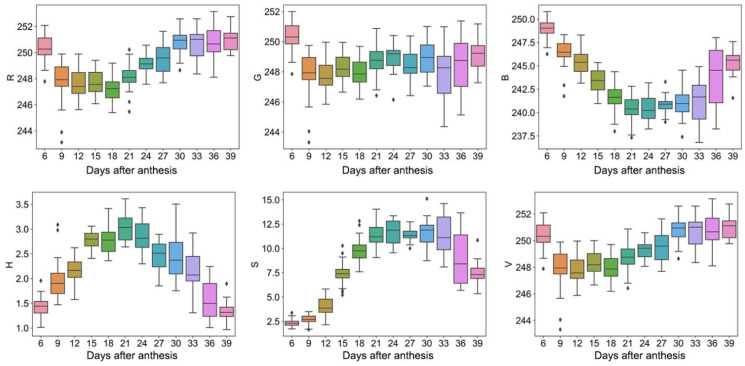
Color traits. Box plots of R, G, B, H, S, and V dynamic after anthesis, where different colored boxes represent different days after anthesis.

**Figure 2 plants-12-04043-f002:**
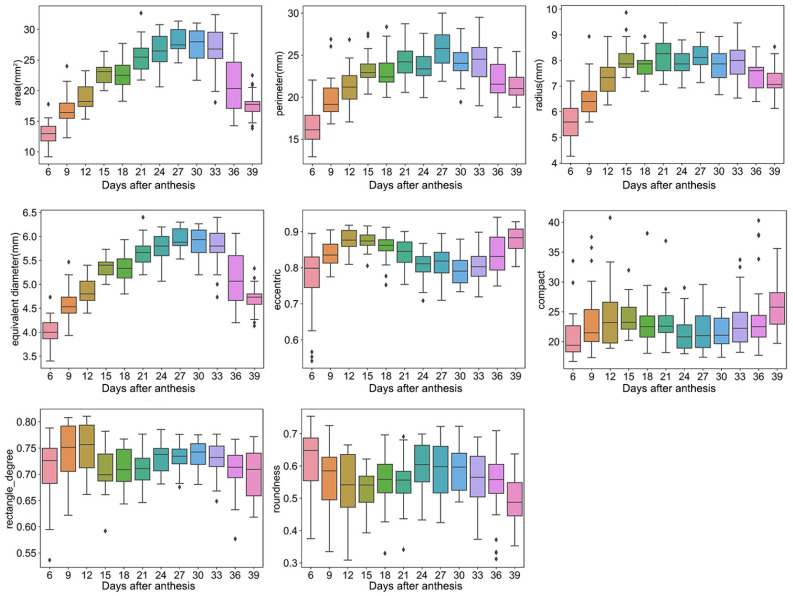
Shape traits. Box plots of area, perimeter, radius, equivalent diameter, eccentric, compact, rectangle degree, and roundness dynamic after anthesis, where different colored boxes represent different days after anthesis.

**Figure 3 plants-12-04043-f003:**
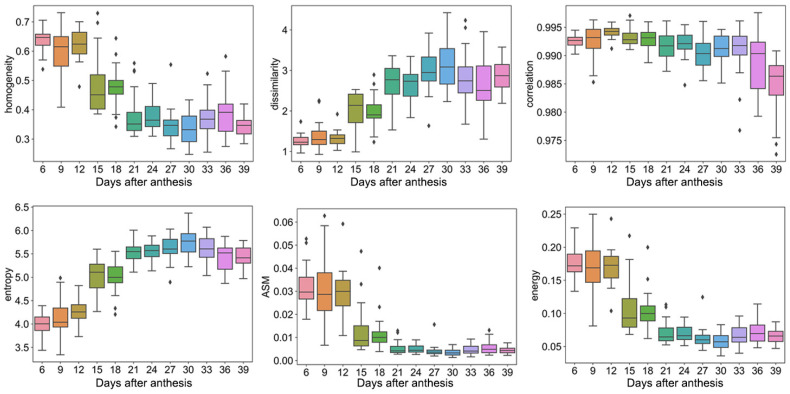
Texture traits. Box plots of homogeneity, dissimilarity, correlation, entropy, ASM, and energy dynamic after anthesis, where different colored boxes represent different days after anthesis.

**Figure 4 plants-12-04043-f004:**
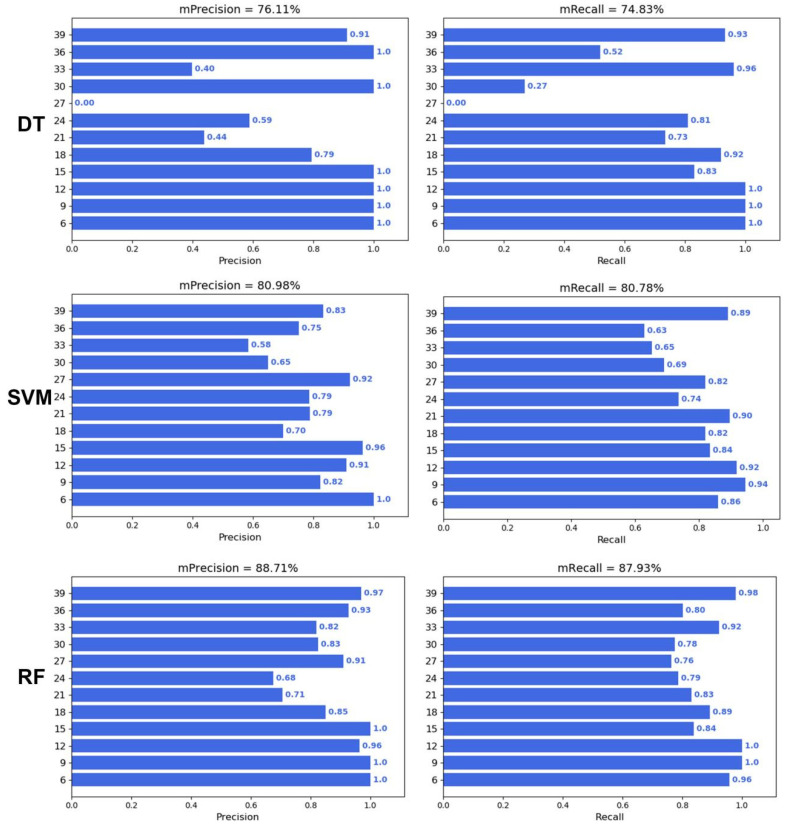
Performance of traditional machine learning.

**Figure 5 plants-12-04043-f005:**
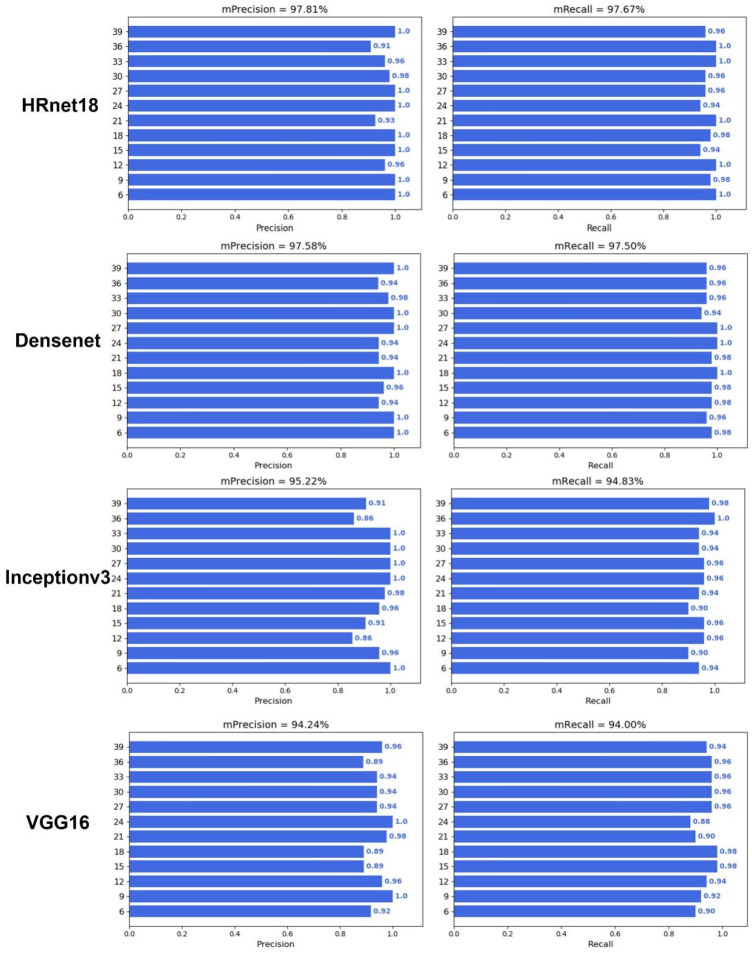
Performance of deep learning with HRnet18, Densenet121, Inception v3, and VGG16.

**Figure 6 plants-12-04043-f006:**
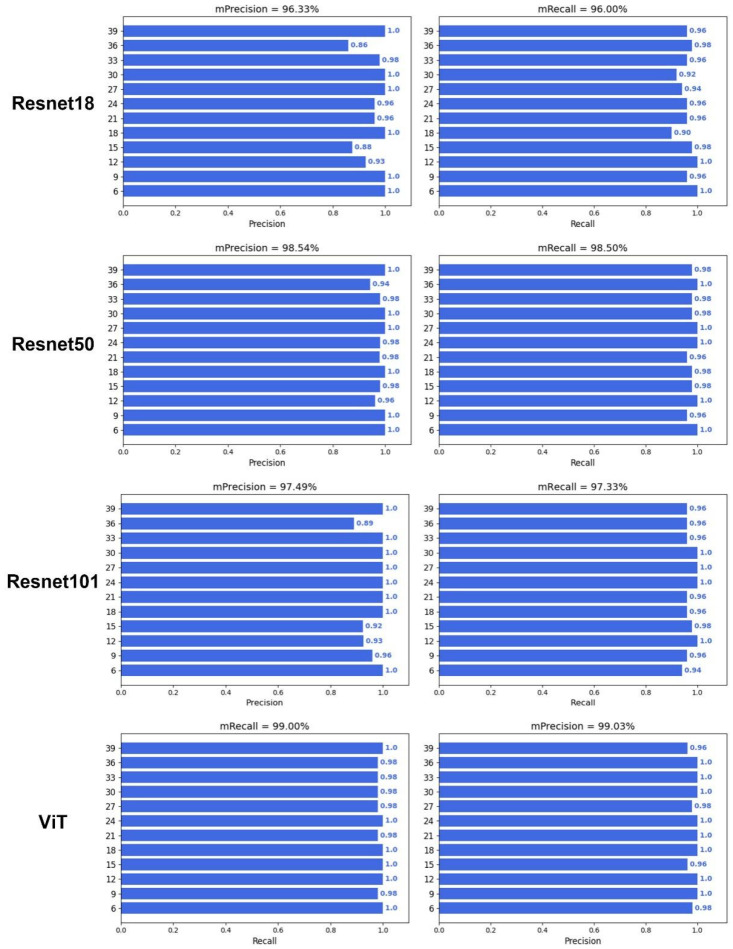
Performance of deep learning with Resnet18, Resnet50, Resnet101, and ViT.

**Figure 7 plants-12-04043-f007:**
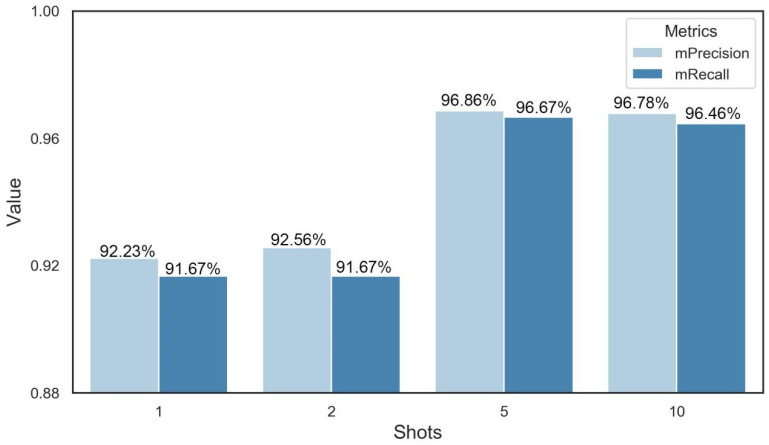
Performance of few-shot learning. The K was set to 1, 2, 5, and 10, which represents the fact that 1, 2, 5, and 10 images were randomly selected from the query dataset.

**Figure 8 plants-12-04043-f008:**
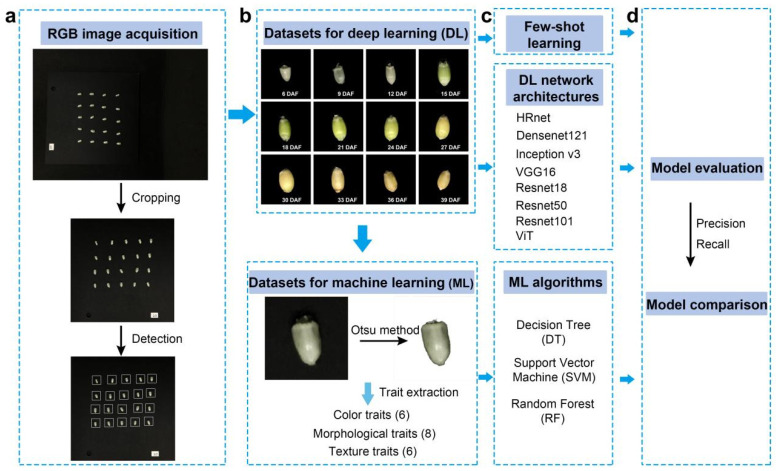
Workflow. (**a**) RGB image acquisition and processing; (**b**) dataset construction; (**c**) model construction; (**d**) model evaluation.

**Table 1 plants-12-04043-t001:** Performance of deep learning algorithms.

Network Architecture	mPrecision	mRecall	F1-Score	Model Size (MB)
HRnet18	0.9781	0.9767	0.9774	74.37
Densenet121	0.9758	0.9750	0.9754	27.15
Inception v3	0.9522	0.9483	0.9502	83.51
VGG 16	0.9424	0.9400	0.9412	512.36
Resnet18	0.9633	0.9600	0.96165	42.73
Resnet50	0.9850	0.9854	0.9852	90.07
Resnet101	0.9749	0.9733	0.9741	162.82
ViT	**0.9903**	**0.9900**	**0.9901**	146.01

Note: Bold numbers represent optimal prediction performance.

## Data Availability

The original contributions presented in the study are included in the article/Appendix A. Further inquiries can be directed to the corresponding authors.

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
