# Peer review of "Comparison of Different Machine Learning Algorithms for the Prediction of the Wheat Grain Filling Stage Using RGB Images"

_plants, 2023, doi:10.3390/plants12234043_

Round 1

Reviewer 1 Report

Comments and Suggestions for Authors

A comparison of different machine learning algorithms for predicting wheat grain-filling stage using RGB images is illustrated in this work and the following are the suggestions or comments for the improvements of this article.

Objectives of this work are presented in the introduction part but I suggest it to convert this to the key contributions of this work by highlighting the novelty of this work.

Since literature survey is not present in this article, it is necessary to strengthen the introduction part by referring to some of the past works done in the similar context.

How image based dataset is used for both ML and Deep Learning algorithms?

How images are classified with approaches like SVM or Decision Trees?

How model size is varying for different approaches?

Is there any model is proposed in this work?

How comparison alone contributes as a research article? is it not a survey?

figure 8 is not clear

MLP is a Ml algorithm not a Deep Learning algorithm?

CNN is used for feature extraction but not for model comparison?

All these technical flaws should be thoroughly corrected for further consideration of this work.

Reviewer 2 Report

Comments and Suggestions for Authors

The manuscript is very clear as to what to study and thus, it reads well. Overall, this manuscript can be published after minor revision suggested below.

-       Please provide a brief explanation on the algorithms tested in sections 3.2 – 3.4.

-       Line 286-292: this paragraph serves better if it is placed in the introduction rather than discussion.

Round 2

Reviewer 1 Report

Comments and Suggestions for Authors

The responses are convincing, and the changes are made according to the review comments. So, this work can be accepted in its present form.